# Rejuvenating the Activity of Usual Antibiotics on Resistant Gram-Negative Bacteria: Recent Issues and Perspectives

**DOI:** 10.3390/ijms24021515

**Published:** 2023-01-12

**Authors:** Jinane Tabcheh, Julia Vergalli, Anne Davin-Régli, Noha Ghanem, Jean-Marie Pages, Charbel Al-Bayssari, Jean Michel Brunel

**Affiliations:** 1Aix Marseille University, INSERM, SSA, MCT, 13385 Marseille, France; 2Faculty of Science 3, Lebanese University, Michel Slayman Tripoli Campus, Tripoli 1352, Lebanon; 3Department of Medical Laboratory Sciences, Faculty of Health Sciences, University of Balamand, Beirut P.O. Box 55251, Lebanon

**Keywords:** multi drug resistance, gram-negative, bacteria, new strategies, antibiotic activity enhancement

## Abstract

Antibiotic resistance continues to evolve and spread beyond all boundaries, resulting in an increase in morbidity and mortality for non-curable infectious diseases. Due to the failure of conventional antimicrobial therapy and the lack of introduction of a novel class of antibiotics, novel strategies have recently emerged to combat these multidrug-resistant infectious microorganisms. In this review, we highlight the development of effective antibiotic combinations and of antibiotics with non-antibiotic activity-enhancing compounds to address the widespread emergence of antibiotic-resistant strains.

## 1. Introduction

The rise of antimicrobial resistance urgently requires the development of new strategies for improving antibiotic activity in infected patients, to minimize the risk of the emergence of resistant strains, and to prevent patients’ reinfection and spreading of resistant bacterial strains in hospitals. Regarding the most concerning bacterial pathogens, the ESKAPE group (*Enterococcus faecium*, *Staphylococcus aureus*, *Klebsiella pneumoniae*, *Acinetobacter baumannii*, *Pseudomonas aeruginosa*, and *Enterobacter* species) represent some of the greatest threats to human health due to their ability to evade antibiotic treatments through various mechanisms of resistance [1]. Thus, it has been recently reported that such infections claimed 5 million lives in the world by 2019, and this trend will only increase in the coming decades with more than 50 million deaths estimated worldwide by 2050 if nothing is done [2]. Despite the human impact of MDR infections, numerous companies have ceased their antibiotic research programs due to the low level of financial profit generated by these products [3]. During recent decades, faced with this worrying health concern, an alternate way has appeared, i.e., rejuvenating usual antibiotics to restore their biological efficacy [4,5,6,7]:By blocking the inactivating enzyme;By tackling the mechanisms that impair the intracellular concentration required for antibacterial activity;By bypassing target mutation.

In this review, we report and discuss the latest strategies presented in the literature to circumvent Gram-negative bacterial resistance. This review will not deal with immunoconjugates or aptamer strategies, since numerous reviews have been devoted to this strategy [8,9,10,11,12,13,14,15].

We focus on papers reporting the new “addressing strategies” to ensure the efficient delivery of antibiotics to internal targets. The use of “cargos” that parasitize the specific transport of nutrients such as siderophores, sugars, or liposomes as vectors is described and discussed to illustrate this original way for repurposing or rejuvenating antibiotics that are inactive against Gram-negative bacterial strains. In this way, the cargo moiety ensures the diffusion across the bacterial outer membrane without exhibiting additional antibacterial activity. This represents per se advantage for preventing a rapid emergence of resistance mechanisms against this new strategy.

Moreover, the development of specific “cargos” based on siderophores and sugars generates an original way by addressing the active drug in defined bacterial compartments, periplasm, or cytoplasm and ensuring significant concentration close to the target.

## 2. Mechanism of Resistance

Antibiotics are natural or synthetic chemicals used to combat bacterial diseases [16]. Antibiotics act in a precise and targeted manner on certain stages of the metabolism of Gram-positive and Gram-negative bacteria. Thus, antibiotics must cross/attack the bacterial cell envelope, the characteristics of which are very different depending on the bacterial species [17]. The general use and misuse of antibiotics favor the emergence of bacterial subpopulations which do not respond positively to treatment; this is the phenomenon of resistance [18]. Gram-positive (+) bacteria possess a cell envelope consisting of a cytoplasmic membrane and a thick wall of peptidoglycan [19], while Gram-negative bacteria present a complex cell envelope consisting of an inner membrane, a periplasm containing a thin wall of peptidoglycan and various cellular molecules, and an outer membrane covered in lipopolysaccharides [20], which provide a more effective barrier than Gram-positive bacteria to antibacterial agents, since they must pass through the two protective bilayers to reach their target. Thus, an efficient antibiotic must fulfill three conditions: (1) it must alter the activity of a target, playing a key biological role in the bacterial cell; (2) it must penetrate to the site of infection and reach its target in a dose-dependent manner; and (3) no or limited side effects must be encountered in the patient’s body when the antibiotics are modified or destroyed, either by the body or by resistant bacteria. Considering the two first points, two types of resistance are described, a natural or an acquired resistance. Some bacterial species are innately resistant towards a specific antibiotic group, and it is an identification marker of the bacterium [21]. For example, the production of a class A beta-lactamase (type SHV-1) by *K. pneumoniae* strongly contributes to penicillin’s resistance (amoxicillin, ticarcillin) [22]. Moreover, natural resistance by impermeability or efflux restricts the spectrum of antibiotic classes (e.g., Macrolides or glycopeptides and most Gram-negative bacilli). Nevertheless, another type of resistance characterized by the sudden emergence of resistance to one or more antibiotics in certain bacteria previously susceptible can occur. In this case, resistance is mainly supported by mutations (in a structural gene or in regulator), derepression of specific genes, or transfers of resistant genes (via mobile element) from a resistant bacterium to a susceptible one; we are speaking of acquired resistance, an epidemiological marker [23].

The clinical major resistance mechanisms are: reduction in internal antibiotic concentration under the required biological threshold (including impermeability and efflux), antibiotic target modification, and antibiotic inactivation (Figure 1) [24].

### 2.1. Membrane Impermeability

In Gram-negative bacteria, porins are transmembrane proteins facilitating the transport of hydrophilic molecules across the outer membrane, while hydrophobic molecules can diffuse through the phospholipid bilayer. Consequently, charged antibiotics, such as ß-lactams, will cross the outer membrane by way of the porin in order to reach the periplasmic target [25]. To limit the antibiotic action, the Gram-negative bacteria have developed mechanisms that decrease their membrane diffusion. Changes in outer membrane permeability impair the influx of hydrophilic antibiotics using porins like β-lactams and fluoroquinolones [26]. An alteration in the porin expression or in porin integrity leads to a decrease in internal concentration and therefore resistance to these antibiotics. For example, *Klebsiella aerogenes* (formerly *Enterobacter*) becomes resistant to cephalosporins because of an amino acid mutation located inside the eyelet of porin that strongly reduces the pore diameter. This structural mutation generates a restricted diffusion of antibiotics through porins, conferring this peculiar resistance [27]. Moreover, it has been recently demonstrated that the porin balance (OmpF vs. OmpC) plays a key role in the penetration of cephalosporins in *Enterobacteriacea* [28].

#### 2.1.1. Antibiotic Target Modification

The antibiotic affinity on a target is a particularly important event during the fight against a bacterial infection. A target modification caused by spontaneous mutation at the gene level (substitutions of amino acids on binding sites located on the target surface), alters the affinity of the antibiotic, favoring the emergence of a resistance level [29]. The involvement of mutations in the QRDR region in a quinolone target, gyrase and topoisomerase enzymes, is extensively documented in Gram-negative pathogens [30]. In addition, mutations of ribosomal proteins are associated with the resistance mechanism exemplified by mutations at uL3 and uL4 or transferable modification of 23S rRNA by methyltransferase enzyme, rendering the Gram-positive bacteria resistant to linezolid [31].

#### 2.1.2. Antibiotic Inactivation

Numerous enzymes produced by bacteria can inactivate the antibiotic either by modifying or hydrolyzing it. Thus, these enzymes can act against β-lactams, aminoglycosides, chloramphenicol, or antibiotics belonging to the macrolide-lincosamide-streptogramin (MLS) family [32]. Inactivation of antibiotics occurs either by hydrolysis or by association with modifying metabolism or elimination (phosphorylation, acetylation, etc.). The β-lactam antibiotics exhibit a β-lactam ring which is the effective part of the molecules, as it inhibits the transpeptidase involved in the synthesis of the Gram-negative bacterial wall; the hydrolysis of this part is one of the degradation mechanisms involved in antibiotic resistance. Thus, the bacterium is able to produce enzymes, termed β-lactamases, able to hydrolyze the β-lactam ring **1** [16,17]. β-lactamases are secreted into the extracellular medium in Gram-positive bacteria and are located in the periplasm in Gram-negative bacteria and involved in the resistance of numerous bacteria towards beta-lactam antibiotics, such as penicillins and cephalosporins [33,34,35,36]. Thus, the amide bond of the β-lactam ring is hydrolyzed, leading to an acyl-enzyme which will then be degraded into an inactive acid. After hydrolysis, the antibiotic is no longer able to bind and inhibit transpeptidase [18,19].

#### 2.1.3. Efflux Pumps

Efflux pumps are transmembrane systems by which the cells expel toxic compounds outside using active transport (energy-dependent), contributing to the bacterial resistance mechanism. A decrease in internal antimicrobial amount under the biological required concentration impairs and limits the antibiotic action [37]. The efflux pumps are classified into six families: the ABC (Adenosine triphosphate-Binding Cassette) family, the MFS (Major Facilitator Superfamily) family, the MATE (Multidrug And Toxic Compound Extrusion) family, the SMR (Small Multidrug Resistance) family, the RND family (Resistance Nodulation cell Division), and finally the PACE family (Proteobacterial Antimicrobial Compounds Efflux) [38]. It is noteworthy that the proton motive force for energy is used by all families of efflux pumps, except the ABC family, which uses the hydrolysis of ATP as the driving force for expelling toxic compounds [24]. The tripartite efflux pumps AcrAB-TolC are extensively reported in the Gram-negative clinical isolates [39].

### 2.2. Strategies to Circumvent Gram-Negative Bacterial Resistance

One of the most promising approaches to circumvent Gram-negative bacterial resistance requires the combination of several types of molecules (e.g., enzyme inhibitor + antibiotic or membrane permeabilizer + antibiotic), for which the biological action can be either additive or synergistic. The additive effect occurs when the observed effect is the sum of the activity of the two partners, while the synergy corresponds to a final action greater than the sum of the activity of the two individual partners [40]. The use of this combination can also minimize the risk of the emergence/spreading of bacterial resistance for the treatment of recurrent infections [41].

#### 2.2.1. Drug Combination

To restore the activity of an antibiotic impaired by a resistance mechanism, an effective strategy consists in associating two antibiotics during patient treatment. The development of the hybrid antibiotic approach needs a chemical covalent connection between two molecules; generating a single hybrid drug that targets bacterial cells can mitigate bacterial resistance mechanisms. In this context, a combination of two antibiotics, such as a fluoroquinolone (ciprofloxacin) and an aminoglycoside (neomycin B), demonstrated a significant delay in resistance emergence compared to each component taken alone, and this is thanks to the multiple antibiotic resistance operon “Mar” involved in the regulation of efflux pump expression and OM permeability [42]. Thus, the authors hypothesized that the ciprofloxacin part of the hybrid molecule inhibits gyrase/topoisomerase enzymes, while the neomycin B part circumvents the effectiveness of the “Mar” operon by inducing a lower mutation rate in the *marR* regulator [43]. It was suggested that chemical structures similar to neomycin B could be replaced in order to stimulate both the permeability of the hybrid and the bypass of the “Mar” response [44].

In the multiresistant (MDR) bacterium *Pseudomonas aeruginosa*, it has been shown that the hybrid tobramycin-moxifloxacin **3** led to better intracellular concentration of antibacterial molecule by (i) increasing the outer membrane permeability and (ii) altering the PMF (proton motive force) which is the energy-driven force of efflux pumps involved in the expulsion of toxic compounds (metabolites, drugs, antibiotics, etc.). Therefore, the combination of moxifloxacin **1** with tobramycin **2** allowed a weaker development of resistance compared to each antibiotic alone (Figure 2) [45]. In addition, the new molecule that associates two patterns could alter the recognition of the antibiotic chemical groups by the affinity sites located inside AcrB pockets, the first step of antibiotic expulsion [39].

Another type of antibiotic used for the treatment of various resistant external infections caused by Gram-negative bacteria is colistin, a poly cationic peptide that disrupts the membrane by interacting with negatively charged components including lipopolysaccharides (LPS). It also has a bactericidal effect through its binding to ribosomes, disruption of bacterial division, alterations in bacterial respiration, production of reactive oxygen species, and induction of structural bonds. The combination of colistin with antimicrobial agents has shown synergism with a wide variety of antimicrobial activities; thus, the combination of colistin–carbapenem is one of the best studied, although synergies have been also observed for colistin associated to quinolones, fosfomycin, or aminoglycosides [46]. Additionally, the combination of colistin with linezolid renders *A. baumannii* susceptible to the latter. A similar effect was observed by using colistin associated with a glycopeptide in a *Galleria Mellonella* model of infection. This antimicrobial effect is achieved due to the disruption of the outer membrane by colistin, which allows linezolid and glycopeptides to enter Gram-negative bacteria that are impermeable to these agents. Additionally, a chloramphenicol–colistin combination appeared to act in synergy in altering efflux pumps against resistant Gram-negative bacteria [47]. It must be noted that the toxicity of colistin limits its clinical use [48].

#### 2.2.2. Antibiotic and β-Lactamase Inhibitor Combination

β-lactam antibiotics used in combination with a β-lactamase inhibitor (Figure 3) constitute a pioneering and efficient strategy to combat β-lactam resistance. Thus, the amoxicillin/clavulanic acid combination, also known as co-amoxiclav or amox-clav and sold under the brand name Augmentin^®^, is a medication used for the treatment of several bacterial infections. Amoxicillin **4** is a semi-synthetic penicillin, and clavulanic acid **7** is a β-lactamase inhibitor, both possessing a similar half-life [49]. They are mainly used for the treatment of acute otitis media [50], lower respiratory tract infections, urinary tract infections, and other infections due to bacteria resistant to amoxicillin alone [51]. The effect of this combination limits the maximum daily dose of amoxicillin administered orally [49], but on the other hand, clavulanic acid has a limited effect on antimicrobial resistance, expands the spectrum of amoxicillin, and therefore results in problems in gut health by increasing side effects [52]. In addition, the sulbactam **8**–ampicillin **5** (or cefoperazone) combination restores the activity of this antibiotic through the β-lactamases’ inhibition and also presents an inherent activity against certain *A. baumannii* strains, which is the advantage for such a combination. However, this combination did not show strong selective pressures for enterobacteria that produce β-lactamases and enterococci that are resistant to vancomycin. It is mainly used for the treatment of urinary, nosocomial, gynecological, and intra-abdominal infections and cellulitis [53]. Furthermore, the therapeutic treatment of sepsis disease combines ampicillin with gentamicin **6** as a first-line antibiotic in children under five years of age, appearing very effective and reducing the mortality rate [54,55]. However, this type of association remains an option of choice. Tazobactam is a member of the class of penicillanic acids known as sulbactam. It is combined with Piperacillin and Ceftolozane for the treatment of a variety of bacterial infections. Piperacillin–tazobactam was approved by the FDA in 1994, and ceftolozane–tazobactam was approved in 2019, providing wider antibacterial coverage for Gram-negative infections and for treating hospital-acquired bacterial pneumonia and ventilator-associated bacterial pneumonia, which are significant causes of morbidity and mortality [56].

Numerous Gram-positive and negative bacteria are resistant to β-lactams due to three potent factors: (1) alteration of the outer membrane in *P. aeruginosa* and *Enterobacteriaceae* species, (2) low affinity of PBPs (penicillin-binding proteins) in *N. gonorrhoeae* and *P. aeruginosa*, (3) destruction of the β-lactam ring by periplasmic β-lactamases in aerobic Gram-negative bacteria, such as *E. coli* and *Klebsiella* species [57]. Combination of this class of antibiotics with β-lactamase inhibitors renders these bacteria more susceptible. Thus, the β-lactamase enzyme is inhibited by β-lactamase inhibitors which bind irreversibly to the enzyme, preventing its action on the β-lactam ring and restoring its antimicrobial activity [58]. *E. cloacae*, *Klebsiella aerogenes*, *C. freundii*, *S. marcescens*, *Providencia stuartii*, *P. aeruginosa*, *Hafnia alvei*, and *Morganella morganii* express chromosomal β-lactamases like AmpC [59], and *E. coli* expresses TEM-1 and derivates, which are the best-known plasmid β-lactamases [60]. β-lactamase inhibitors are active on both types.

Carbapenem resistance is due to multiple resistance mechanisms, such as mutations of porins, efflux pumps, or carbapenemase (hydrolysing enzymes). Vaborbactam is a non-β-lactam β-lactamase inhibitor; its association with meropenem (a carbapenem) inhibits class A serine carbapenemases, thus restoring the activity of meropenem. This association shows excellent activity against Gram-negative bacteria in vitro and specifically against *K. pneumoniae* (KPC), which is the most resistant *Enterobacteriaceae* to carbapenemases. A study on an *E. coli* clinical strain reported a lower MIC value with meropenem–vaborbactam compared to the combination with tazobactam and clavulanic acid (*β-lactamase* inhibitors) [61,62].

Another non-β-lactam β-lactamase inhibitor is avibactam, which can irreversibly acylate class A, class C, and class D β-lactamases. This compound presents a low molecular weight, long half-life, polarity, and an interaction with residues near the active sites of β-lactamases. Ceftazidime, belonging to the third generation of the cephalosporin group of β-lactam antibiotics, binds to penicillin-binding protein (PBP) of Gram-negative bacilli and inhibits cell wall synthesis [63]. Its association with avibactam has shown great therapeutic efficacy in patients suffering from complicated urinary tract infections (ICIA) and nosocomial pneumonia [64]. It must be noted that regarding these enzymatic inhibitors, their activity can be altered if the outer membrane permeability is decreased following lack of porin or other membrane modifications [39].

#### 2.2.3. Efflux Pumps Inhibitors

Efflux pumps expel the antibiotics towards the extracellular environment and contribute to bacterial resistance. Some are selective and only expel specific substrates, while others are non-selective and expel various classes of antibiotics and non-specific substrates (detergents, organic solvents, dyes, etc.) [39]. In this case, a possible strategy consists of blocking efflux systems by so-called efflux pump inhibitors (EPIs) and therefore restoring the susceptibility of resistant strains. Several methods are used to target efflux pumps: (1) inhibit the expression of genes encoding these pumps, (2) deplete the pumps of the energy necessary for their functioning, (3) compete in the inner membrane transporter, (4) block the outlet channel in the outer membrane, (5) prevent the assembly of pump components at the membrane level [24]. For example, the use of a natural steroidal alkaloid extracted from the antidysentery plant *Holarrhena*, conessine **11**, demonstrated a reduction in the minimum inhibitory concentrations of various antibiotics including tetracycline, cefotaxime, levofloxacin, novobiocin, rifampicin, and erythromycin and an inhibitory activity of MexAB-OprM efflux pumps (Figure 4) [65]. Epigallocatechin-3-gallate (EGCG **12**) extracted from tea is another compound responsible for inhibiting this type of efflux pump in *P. aeruginosa* [66].

Reserpine **13** is an antipsychotic drug extracted from the alkaloid plants *Rauwolfia serpentina* targeting the efflux pumps of the superfamilies MFS and RND by directly interacting with amino acid residues in the efflux transport protein of tetracycline antibiotic. This plays another role against *S. aureus* by increasing 4-fold the activity of norfloxacin released by the NorA pump. Piperine **14,** which is another alkaloid, is able to inhibit this NorA efflux pump and then enhance ciprofloxacin accumulation in *S. aureus* [67].

Quinoline derivatives such as pyridoquinolones **15** act as competitive inhibitors of the AcrAB-TolC efflux pump and restore norfloxacin activity against *P. aeruginosa*. Additionally, the amide dipeptide compound PaβN **16** is one of the first EPIs discovered through a chemical approach. It inhibits the efflux pumps of the RND family and increases the activity of many antibiotics such as fluoroquinolones, macrolides, and phenicols, but on the other hand, it presents a toxicity towards mammalian cells, suggesting a limited therapeutic use [68,69].

#### 2.2.4. Liposomes Addressing Resistance

Liposomes are vesicles formed by a phospholipid bilayer which mimic biological membranes. The surface properties of liposomes are determined by their polar (hydrophilic) heads, and the membrane fluidity is determined by their non-polar (hydrophobic) tails, which can organize the diffusion of hydrophilic and hydrophobic compounds due to hydrophilic heads and hydrophobic tails [70]. The classification of liposomes depends on their sizes: unilamellar vesicles (a single phospholipid bilayer), which are classified into (1) large vesicles (LUV) or (2) small unilamellar vesicles (SUV), and multilamellar vesicles (several phospholipid bilayers) [71]. Loading of liposomes with antibiotics makes several Gram-positive and Gram-negative bacteria susceptible to specific antibiotics [72,73,74].

The liposomes adhere to the bacterial membrane (outer membrane) and contribute to the release of the antibiotic into the periplasmic space (Figure 5) [41]. The comparison between the Minimum Inhibitory Concentration (MIC) values of the antibiotic (tetracycline) present in a solution and in a liposome shows an important decrease (MIC in the solution is equal to 0.063 and 0.125 μg/mL for *S. aureus* and *S. epidermidis*, respectively, whereas there is a 3.9-fold decrease for *S. aureus* and 12.8-fold for *S. epidermidis* when the antibiotic is present in the liposome). This reduction indicates that the liposomes decrease the antibiotic doses required due to the fusion of the liposome structures with the bacterial membranes according to the mimetic characteristics [37,38].

#### 2.2.5. Potentiation of an Antibiotic Activity in the Presence of Natural Organic Compounds

Natural organic compounds, such as flavonoid polyphenol curcumin **17,** can also be used to potentialize antibiotic antibacterial activity (Figure 6). Thus, this led to the restoration of the bactericidal activity of oxacillin and norfloxacin against resistant *S. aureus* (MRSA) and linezolid activity against *Mycobacterium abscessus* by permeabilizing the membrane, altering cell viability and thus facilitating the penetration of the antibiotic. It is used for the treatment of tory bowel syndrome, diabetes, and asthma in Asian medicine [75].

On the other hand, the combination of mammalian proteins, such as lactoferrin with iron, an essential component for cell proliferation, allows for the restoration of the activity of oxacillin against MRSA (synergistic effect) by disrupting the cell membrane and decreasing the MIC value [76]. Essential oils are natural biological products used in medical therapy, mainly dermatology, to restore antibiotic activity. They deal with the inhibition of efflux pumps but also with the inhibition of biochemical pathways, protective enzymes, and membranes, in the case of thyme oil. The use of isoeugenol induces a reduction in MIC and a synergy by association with ciprofloxacin in resistant *S. aureus*. It was able to alter these three bacterial pumps but presented toxicity in *Drosophila melanogaster* for an acute 24-h period [77]. Furthermore, carvacrol **18** acts by destabilizing the bacterial membrane, allowing restoration of the activity of cefixime and pristamycin against *P. aeruginosa*. On the other hand, the activity of sarafloxacin and florfenicol is restored in the presence of oregano against the β-lactamase produced by *E. coli* [78].

#### 2.2.6. Potentiation of an Antibiotic Activity in the Presence of Inorganic Compounds

Inorganic compounds including metals, ceramics, glass ceramics, and polymers can act as antimicrobial agents through several mechanisms: alteration of membrane function, production of reactive oxygen species (ROS), release of toxic ions, and loss of oxygen enzyme activity. Their size, shape and concentration can have a great impact on their antimicrobial activity [41].

These size- and structure-dependent properties rely on the ability to synthesize and modify materials below 100 nm, ranging up to 200 nm for use in medicine and pharmaceutical products, such as metal nanoparticles (NPs) and metal oxide nanoparticles; NPs are useful due to their chemical and mechanical stability, control of morphology, and particle size [79]. For example, treatment of *E. coli* with silver nanoparticles (AgNPs) resulted in the formation of “pits” in the bacterial cell walls and subsequent death of the bacteria. An increase in antimicrobial activity of this strain is shown by using small AgNP particles with a larger surface/volume ratio. Therefore, it is clear that small particles are more effective than large ones [80].

AgNPs are involved in three mechanisms to inhibit bacterial physiology, as illustrated in Figure 7.

The first acts on membrane permeabilization: AgNPs penetrate the periplasmic space, and its adhesion to the internal membrane induces the destabilization of the bacterial cell and leakage of the cell content, leading to the death of the bacteria.

The second mechanism deals with the affinity between AgNPs and sulfur or phosphorus present in the intracellular medium at the level of DNA and proteins; an alteration of the respiratory chain present in the inner membrane could occur due to the affinity of AgNPs towards thiol groups of enzymes, inducing the release of reactive oxygen species (ROS), resulting in damage of the cellular machinery and activation of the apoptotic pathway.

The third mechanism consists in the interaction of the silver released by the nanoparticles with cellular components, which can lead to the modification of the metabolic pathways and alteration of the membrane and of the genetic material [81].

In this context, enhancement of vancomycin and ampicillin activity against *K. pneumoniae* and *E. coli* has been encountered by grafting them to silver nanoparticles [82].

Several methods are used to synthesize such nanoparticles: physical, chemical, and biological. The physical method consists of evaporating and condensing using a tube furnace to obtain nanoparticles. This method is fast and does not use any dangerous chemicals, but on the other hand, there is a very high energy consumption with a low production yield and contamination with solvents.

The chemical method consists of reducing the silver salts by nucleation followed by subsequent growth. This has a high yield, ease of production of AgNPs, and a low cost, but the disadvantage is the use of chemicals harmful to human health [80]. Finally, the biological method consists of the use of reducing agents extracted from plants, such as the leaf of *Artemisia argvi*, which has a strong antioxidant activity and a biosynthesis potential for phenol and flavonoid compounds, which allows it to reduce Ag^+^ to AgNPs [83]. The use of AgNPs in combination with an antibiotic has the advantage of reducing the necessary doses of the antibiotic, which reduces possible side effects [84]. The nanoparticles can also be used as carriers by forming complexes which allow their release and their selectivity [47,50].

In another field, dressing is used to heal severe wounds, but by contamination, it impedes healing due to the competition between cells and bacteria on the wound surfaces. For a wound dressing to be ideal, it must preserve cell attachment to regenerate tissue and have an antibacterial effect. The use of methylcellulose (MC) foams alone against *E. coli* and *S. aureus* has no effect on their growth; however, its association with Manuka honey (MH) MC-MH produced by freeze-drying shows a clear reduction in growth. Meanwhile their association with B3 (dense borate ion), based on bioactive glass (BG) doped with copper (MC-MH-B3-BG-Cu), shows a much more effective antimicrobial effect compared to MC-MH, the bacterial viability of *E. coli* decreases from 60% to 0%. Copper is an antibacterial ion that penetrates the bacterium and leads to the degradation of its DNA. Therefore, MC-MH-B3-BG-Cu foams can be used as a wound dressing for the treatment of infected wounds [85].

The delivery of drugs can also be carried out using polymer NPs prepared from biodegradable and biocompatible synthetic polymers. The most widely used is poly-lactic-co-glycolic acid (PLGA). It is based on the electrostatic interaction between the opposite charges of the drug with the polyelectrolytes. Several parameters are important for the use of polymers, such as their ease of preparation and functionalization, their numerous structural integrities, their stability during storage, the ability to control them during drug release by selecting the polymer type, their composition, and their molecular weight [86].

Among studies, the most interesting polymer to target a drug is chitosan (ChNP), which has the effect of preventing infections and fighting against microbial agents. Chitosan is a cationic polysaccharide derived from chitin formed from N-acetylglucosamine and D-glucosamine which has non-toxic, biocompatible, biodegradable, and low-allergy properties, which could signal its usefulness as a versatile biopolymer [87]. The mode of action of chitosan is based on its positive charge, which promotes a very close interaction with the bacterial wall formed of peptidoglycan rich in carboxyl and negatively charged amine; this increases the adhesive potential of chitosan. In addition, if its molecular weight is lower, it enters into the bacterial cell and acts by inhibiting RNA synthesis and bacterial proteins by binding to DNA and improves the properties of permeation and inhibition of efflux pumps [88].

## 3. Outer Membrane Permeabilization

The outer membrane (OM) of Gram-negative bacteria constitutes an additional permeability barrier which protects the cell from many more antibacterial agents compared to Gram-positive bacteria. The negatively charged components of the OM interact with permeabilizers, promoting the entry of impermeable molecules. Permeabilizers are considered as potential antibiotic adjuvants to fight against bacterial resistance [39,89]. Pentamidine, an anti-protozoan drug, has shown effective activity against a wide range of Gram-negative bacteria by interacting with the lipopolysaccharide, and as a result, disrupting the outer membrane. In addition, the cationic peptide and its derivatives destroy the outer membrane by electrostatic interaction. For example, *P. aeruginosa* is made susceptible by using a short proline-rich cationic lipopeptide which potentiates minocycline and rifampicin activities [90].

Benzalkonium chloride (BAC) at different concentrations affects the OM and then permeabilizes the inner membrane, but it must be used at a lower concentration than the MIC because a sublethal concentration leads to bacterial tolerance and resistance to antibiotics [91].

Menadione, a soluble synthetic vitamin converted into vitamin K2 at the intestinal level, was used to see its effect on membrane permeability against multiresistant bacteria such as *S. aureus*, *P. aeruginosa*, and *E. coli*. It has been demonstrated to have an antibacterial activity only against the *P. aeruginosa* bacterium, but when combined with antibiotics of the aminoglycoside family, it produces a synergistic effect and decreases the MICs for these antibiotics [92]. Otherwise, AMPs (antimicrobial peptides) possess an antibacterial activity by altering membrane barrier function by forming pores. Thus, nisin, an antibiotic that plays a role in the formation of pores in the membrane of sensitive bacteria, led to a rapid efflux of small molecules from the cells and then to a dissipation of the membrane potential. This antibiotic uses lipid-bound cell envelope precursors like lipid II [93]. Polymyxin B is a polycationic peptide antibiotic that also disrupts the bacterial membrane by binding to lipopolysaccharides (LPS), rendering this membrane permeable to various antibiotics and allowing them to enter the bacterial cell when the permeability of the OM is altered [39,94].

Polyamines (Figure 8) also have a role in the fight against bacterial resistance. The two quinoline polyamine derivatives **19** and **20** are used as an adjuvant for doxycycline by improving its activity against *P. aeruginosa*. A disruption of integrity and depolarization of bacterial membranes have been observed with derivative **20,** while the mechanism of action of derivative **19** is not yet known [95]. The polyamine farnesyl **21** to **24** derivatives make the bacterium *P. aeruginosa* sensitive to the antibiotic tetracycline, and this is linked to the hydrophobicity of the antibiotic and to the alteration of the integrity of the bacterial outer membrane [96,97]. In addition, the polyamine derivatives motuporamine **25** and **26,** extracted from the marine sponge *Xestospongia exigua*, are used as adjuvants for doxycycline; they have an antibacterial effect by modifying the transmembrane electrical potential, which results in an alteration of proton hemostasis [98].

## 4. Cargo Delivery: Conjugation of Antibiotics with Siderophores or Sugars (The “Trojan Horse” Strategy)

The “Trojan horse concept” emerged from studies on specific systems involved in the capture and selective transport through bacterial membranes of iron, carbohydrates, and essential compounds to bacterial physiology [99]. The aim is to synthesize a vector based on a cargo (siderophore, sugar) fused to an antibiotic in order to efficiently direct the agent to the internal bacterial target by hijacking the respective transport systems.

### 4.1. Siderophores

Siderophores are low-molecular-weight metabolites produced by microorganisms and plants involved in iron capture. They have a strong affinity for iron, compete with iron transporters in the host body, and belong to pathogen virulence factors [64,65]. When the availability of iron is limited in the environment, bacteria synthesize and secrete siderophores to scavenge iron and transport it inside the bacterial cell. Two types of siderophores exist: endogenous siderophores synthesized by the bacterium itself and exogenous siderophores synthesized by other microorganisms “cross-feeding” [100,101,102].

*Pyoverdins* are mixed, fluorescent siderophores produced by *Pseudomonaceae* and have a very complex structure and high molecular weight. The synthesis of pyoverdins begins in the cytoplasm, and then they are secreted by the PvdRT-OpmQ pump into the extracellular medium [103,104].

*Pyocholine* pertains to a second class of siderophore synthesized by *P. aeruginosa* which has a lower affinity for iron than pyoverdine. The synthesis of pyocholine takes place in the cytoplasm [69,70].

As previously mentioned, bacteria can also use exogenous siderophores synthesized by another bacterial species to transport iron to the cytoplasm, such as *Enterobactin* which is the main siderophore produced by Gram-negative bacteria; in particular *E. coli* is used by *P. aeruginosa* [105,106]. The transport of Enterobactin in *P. aeruginosa* occurs across the outer membrane by two transporters Pfi A and PrA; however, its transport across the inner membrane still remains unclear [107,108].

The use of iron transporters (siderophores) could be an interesting alternative to circumvent the obstacle presented by the Gram-negative bacteria envelope during the entry of many antibiotics. Thus, the construction of antibiotics linked to siderophores can facilitate the penetration of the antibiotic in the cytoplasm. Gram-negative bacteria import Fe3+ in the form of ferric siderophore complexes across the outer membrane by TonB-dependent transporters (TBDTs). This translocation depends on the complex composed of the ExB/ExbD/TonB proteins present in the bacterial membrane [109]. Generally, a subset of siderophores from the same chemical family is recognized by a single TBDT. Once the ferric siderophore complex is present within periplasmic space, it is recognized by periplasmic binding proteins, which allows it to cross the inner membrane by transporters involved in the uptake of ferric siderophores. The ferric siderophore complex must be dissociated by several mechanisms: (1) hydrolysis of the siderophores by esterase, (2) modification of the siderophores such as acetylation, (3) release of proton iron [110].

Siderophores can be used as vectors, allowing the vectorized antibiotic to easily pass through the bacterial membrane, reach the target, and therefore reduce the phenomenon of resistance. This strategy consists of connecting a linker between the siderophore and the antibiotic (Figure 9). This linker must be variable in length and chemical properties depending on the target, stable in the extracellular environment, and hydrolysable in the bacterial cell [111].

Albomycins are sideromycins secreted by the bacterium *Streptomyces*; their siderophore part is composed of ferrichrome, the serine is the linker, and the antibiotic is composed of a thioribosyl-pyrimidine which inhibits the enzyme seryl t-RNA synthetase and therefore the protein synthesis [112]. In *E. coli*, peptidase N cleaves the linker between the siderophore and the antibiotic. If the pepN gene coding for this peptidase is mutated, the albomycin loses it biological activity and transports iron into the bacterial cell. It is used in the treatment of various bacterial infections, so this strategy will help the use of albomycins as new antimicrobial agents, even if it is more effective against Gram-negative bacteria than Gram-positive bacteria [113].

On the other hand, microcins are also sideromycins synthesized from *K. pneumoniae* Ryc 492, whose siderophore component is composed of enterobactin with an antibiotic, which is a peptide linked by a glycosidic group. The transport of microcin into the cytoplasm results in the cleavage of the ester group linking the sugar with the antibiotic by intracellular hydrolases [114].

The released peptide leads to the depolarization of the bacterial membrane by forming pores. The antibiotic activity increases (10-fold) when the vectorization is made by a catecholated siderophore [115].

Interestingly, pyoverdine has an amine group in its structure, which allows it to bind to the antibiotic via a linker; the antibiotic is ampicillin, which is a β-lactam. Conjugates **27** and **28** (Figure 10) tested against ampicillin-resistant *P. aeruginosa* ATCC27853 and ATCC13692 have reported high antibacterial activities, with MICs ranging from 0.01 pg/mL to 0.67 pg/mL, respectively. The pyoverdine–ampicillin conjugate is dissociated from the iron moiety in the periplasm, and the antibiotic can inhibit peptidoglycan synthesis. Its fusion with fluoroquinolone does not generate any activity, since pyoverdine has a periplasmic location, while the target of the fluoroquinolone is cytoplasmic [115].

Pyochelin is also used in the Trojan horse strategy as a cargo approach (Figure 11). C5 of pyochelin is modified, and then two analogs are synthesized, **29** having a triple bond and **30** possessing a single bond (softer aliphatic chain) [115]. The conjugation of the terminal amine group of pyochelin with the antibiotic norfloxacin against *P. aeruginosa* ATCC1592 cultured in succinate medium (starved in iron level) shows that the conjugates **31** and **32** bearing a non-hydrolysable succinic linker do not present any antimicrobial activity at a concentration of 10 μM. However, this conjugation, made this time by a hydrolysable linker **33** and **34** at the same concentration, shows a high antimicrobial activity and a high inhibition of bacterial growth. Thus, the nature of the linker involved in the design of these conjugates seems to be important for the delivery of the antibiotic to its target and induces an effective activity [115].

Nowadays, a highly targeted Trojan horse strategy has emerged with cefiderocol **35** (Figure 12). Its structure contains a catechol-type siderophore fused to antibiotic “cephalosporin”, this hybrid being active against Gram-negative resistant bacteria like *P. aeruginosa*, *A. baumannii Enterobacteriaceae*, and *S. maltophilia* [116]. It has received approval for use from the United States Food and Drug Administration (FDA). In the structure of cefiderocol, ceftazidine, and cefepime, two cephalosporins can be distinguished by the presence of a catechol group in position 3 of the side chain, which allows it to form a chelated complex with iron and therefore become a carrier in the bacterial cell using the iron transport system (the C-3 side chain prevents recognition and degradation by β-lactamases, and the C-7 side chain enhances stability against β-lactamases) [117].

Once cefiderocol is in the periplasm, iron is released, and the antibiotic binds to penicillin-binding proteins (PBPs) and then inhibits peptidoglycan synthesis, resulting in cell death (Figure 13). It is stable against ESBL β-lactamases (extended-spectrum β-lactam), AmpC, and carbapenemases and active against class A, B, C, and D β-lactamases. Its activity against *A. baumannii* is more potent than ceftazidime–avibactam and meropenem against enterobacteria-producing carbapenemases (KPC) resistant to meropenem. This activity is the same or slightly higher than ceftazidime–avibactam against *K. pneumoniae* [118].

The penetration of cefiderocol into bacterial cells is independent of OM porins, which allows it to remain active when the bacterium is resistant to β-lactams by lack of porin. In addition, since it is mainly excreted by the kidneys, it is necessary to finely adjust the patient therapy. Cefiderocol is recommended for the treatment of complicated urinary tract infections (CUTI) such as pyelonephritis, and its use is being considered for the treatment of nosocomial pneumonia, bloodstream infections, and carbapenem-resistant infections [119].

### 4.2. MaltoCargo Alternatives

A different strategy consists of using maltodextrin–cargo conjugates to circumvent the antibiotic resistance of Gram-negative bacteria [120]. Thus, these conjugates were designed to cross the outer membrane via the porin LamB channel involved in maltodextrin transport, resulting in an accumulation in the periplasm. Then, MalE, the periplasmic binding protein, captures the maltodextrin moiety and allows its transport to the cytoplasm via specific transporters inserted in the inner membrane (MalF,G,K). Finally, maltodextrin is metabolized by the MalP enzyme and allows the release of the antibiotic into the cytoplasm [121]. The interest and advantage of maltotriose conjugate are that it can reach the periplasm and the cytoplasm at important internal concentrations, activating the maltose regulon which results in the auto-induction of its own entry pathway, representing an interesting prototype for releasing molecules into the cytoplasm of Gram-negative bacteria (Figure 14) [122,123].

## 5. Conclusions

A large panel of approaches has been studied to bypass bacterial resistance mechanisms, including modifying the target of the antibiotic, inhibiting degradative enzymes, blocking efflux pumps, permeabilizing the bacterial membrane, and hijacking selective transport systems. Antibacterial activity is improved either by combination of the antibiotic with drugs or with β-lactamase inhibitors, by addressing liposomes loaded with antibiotics, or by combination with organic and inorganic compounds. In addition, the Trojan horse strategy has also been developed to combat bacterial resistance; siderophore or maltose are linked to the antibiotic molecule and used as a cargo to deliver the antibiotics into the bacterial cell.

Until now, regarding bacterial resistance towards antibiotics, an important challenge remains the determination of intracellular concentration of active molecules close to the target [39,124]; this parameter is highly important, since the MIC assay give only an indirect and rough estimation of the in situ antibiotic activity after several hours of incubation in vitro. Recently, some smart studies using fluorimetry and mass spectrometry, in vitro and in cellulo, have quantified the diffusion rate and the internal concentrations of several antibiotics in bacterial strains expressing various resistance mechanisms [39,89]. The relationships between the expression level of membrane transporters, their selectivity for the antibiotic globularity and charges, their affinity for pharmacophoric groups located on the antibiotic surface, and their susceptibility to certain blockers must be precisely analyzed for each bacteria species. Thus, the concepts SICAR and RTC2T now allow the comparison of the biological effect of adjuvants/combinations/cargos in the efficacy of diffusion/transport across bacterial membranes and intra-bacterial accumulation of several antibiotics.

In addition, with these recent tools, it will be possible to characterize the efficient ratio of antibacterial agent–adjuvant/cargo (enhancer); this key point will precisely determine the true role of the partner in the antibiotic activity and the molecular level of active restoration that depends on the antibiotic chemical group, location of internal target, the type of resistance mechanism, the bacterial envelope, etc.

Moreover, the selection of appropriate “cargo” can be done according to the location of the target in a defined bacterial compartment, an interesting point that allows the improvement of the kinetics of antibiotic distribution in the bacterial cell. This key advantage is illustrated in the case of “SiderophoreCargos” that address the antibiotic to periplasmic space. Interestingly, “MaltoCargos” are able to deliver antibiotic by hijacking not only outer the membrane channel (maltoporin) that provides the outer membrane diffusion, but also the maltose permeases involved in the transport across the inner membrane. This consequently achieves a significant concentration of the drug in the cytoplasm. This choice of smart “cargo” paves the way to specifically deliver the active agent to a selected target located in a defined bacterial compartment.

Finally, these perspectives allow a rational chemical design of adjuvants to evaluate the efficiency of different strategies and to prepare the future collection of antibiotic partner treatments.

## Figures and Tables

**Figure 1 ijms-24-01515-f001:**
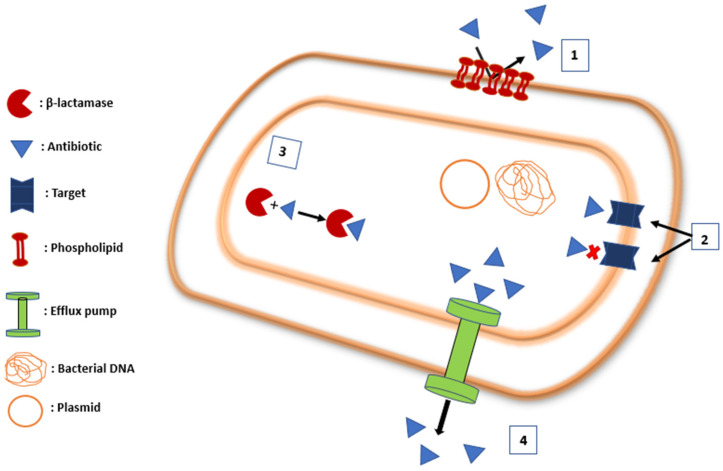
Potent mechanism of resistance of a bacterium. (1) Membrane impermeability, (2) Target modification, (3) Antibiotic inactivation, (4) Efflux pumps.

**Figure 2 ijms-24-01515-f002:**
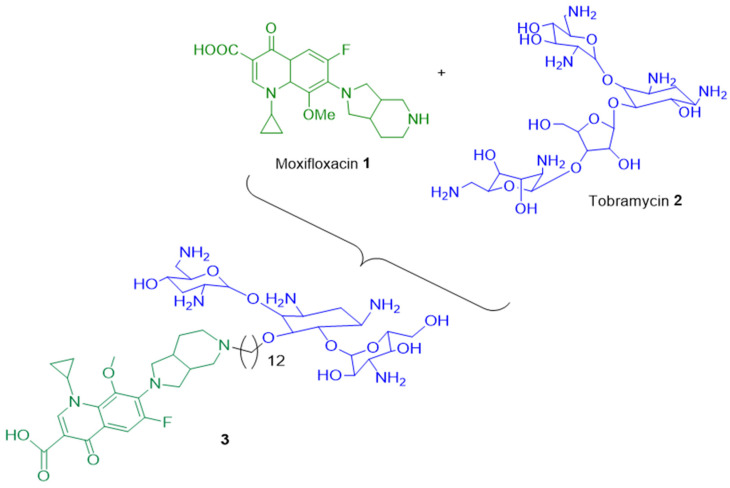
Drug combination **3** of moxifloxacin with tobramycin.

**Figure 3 ijms-24-01515-f003:**
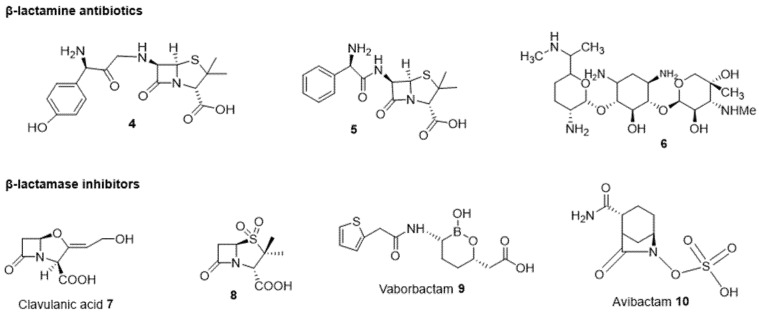
Structure of β-lactam antibiotics **4–6** and β-lactamase inhibitors **7–10**.

**Figure 4 ijms-24-01515-f004:**
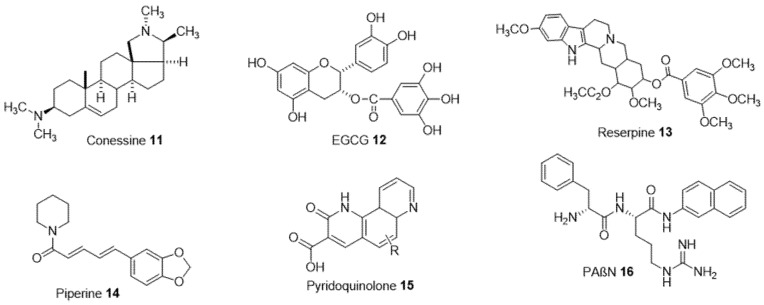
Structure of efflux pumps inhibitors **11–16**.

**Figure 5 ijms-24-01515-f005:**
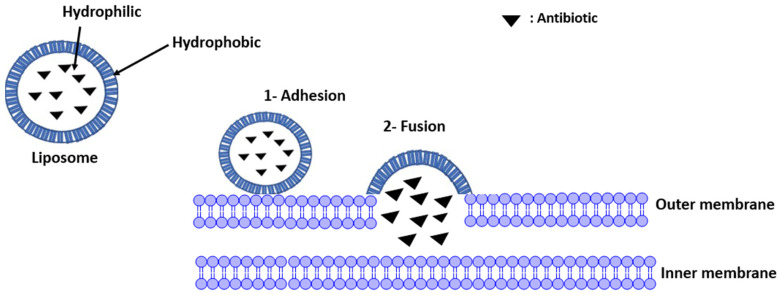
Fusion of liposomes to microbial cell membranes and release of antibiotic.

**Figure 6 ijms-24-01515-f006:**
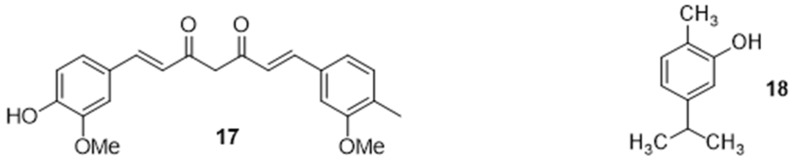
Structure of curcumin **17** and carvacrol **18**.

**Figure 7 ijms-24-01515-f007:**
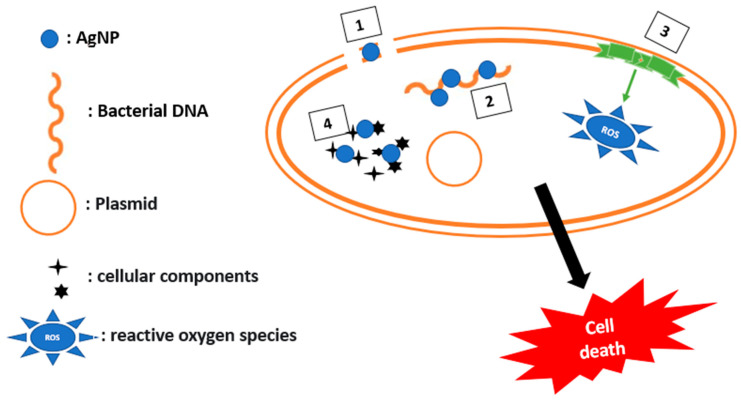
Mechanisms to alter bacterial physiology by using AgNPs. (1) Membrane permeabilization, (2) Interaction with DNA, (3) Liberation of ROS, (4) Interaction with cellular components.

**Figure 8 ijms-24-01515-f008:**
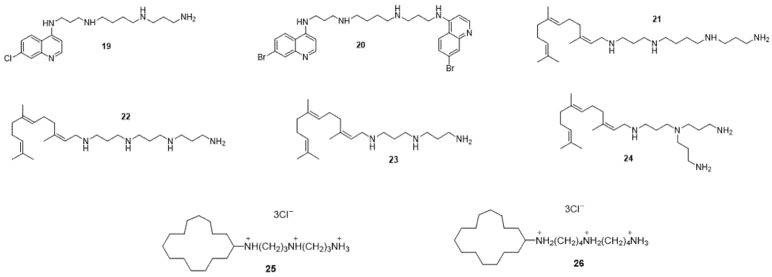
Polyamine derivatives **19–26**.

**Figure 9 ijms-24-01515-f009:**
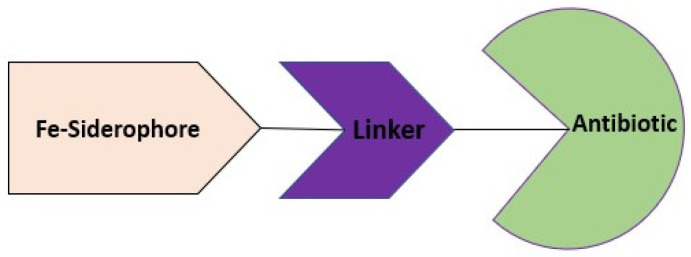
Antibiotics conjugated with siderophores.

**Figure 10 ijms-24-01515-f010:**
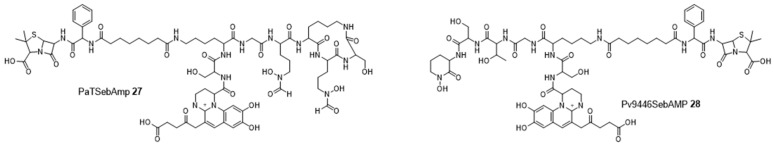
Pyoverdins conjugates **27–28**.

**Figure 11 ijms-24-01515-f011:**
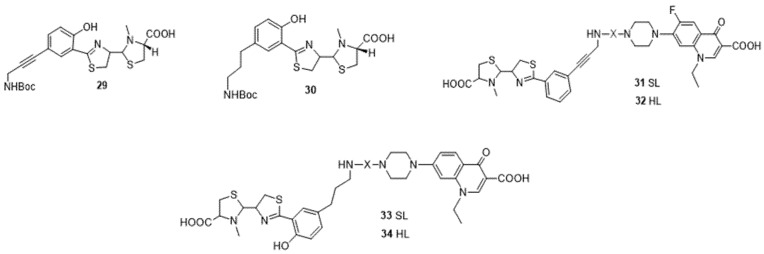
Pyochelin conjugates **31–34**.

**Figure 12 ijms-24-01515-f012:**
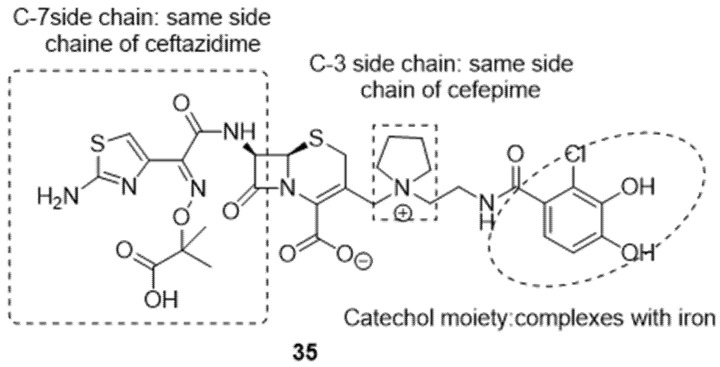
Structure of cefiderocol **35**.

**Figure 13 ijms-24-01515-f013:**
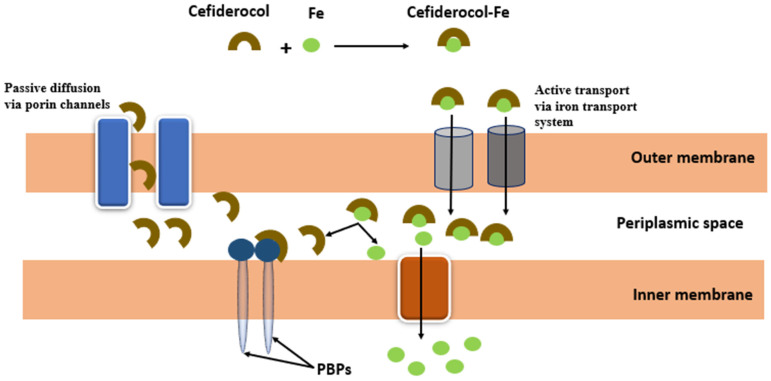
Mechanism of action of cefiderocol **35**.

**Figure 14 ijms-24-01515-f014:**
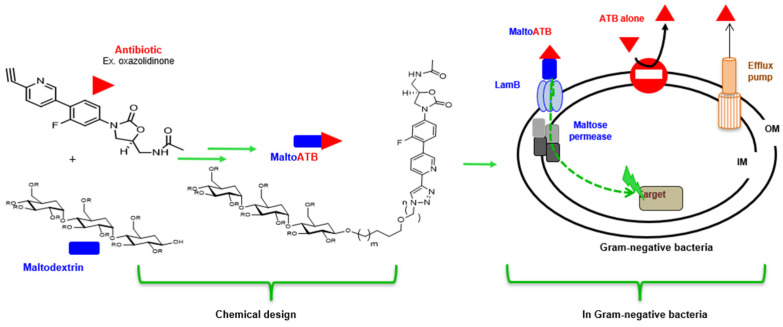
Maltocargo strategy conjugates to circumvent the antibiotic resistance of Gram-negative bacteria.

## Data Availability

Not applicable.

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
