# Peer review of "Rejuvenating the Activity of Usual Antibiotics on Resistant Gram-Negative Bacteria: Recent Issues and Perspectives"

_ijms, 2023, doi:10.3390/ijms24021515_

Round 1

Reviewer 1 Report

Thios review is well done, fluent and the comprehension is optimal.

Some English mistakes and minor revision throughout the entire manuscript.

Author Response

Dear reviewer 

According to your recommendation we have read the text carefully and corrected some english mistakes. 

Reviewer 2 Report

The manuscript "Rejuvenating the activity of usual antibiotics on resistant
Gram-negative bacteria, recent issues and perspectives" summarizes the literature on various approaches to overcoming resistance of Gram-negative bacteria to known antibacterials, both natural and acquired. Although the manuscript deals with an interesting and relevant topic, it has significant flaws.

Major:

1. First, there are no references in the manuscript to other review publications, either on the main topic of this review or on specific sections. Meanwhile, even quick search reveals numerous review publications in this field (e.g. 10.1080/14787210.2020.1705155, 10.1038/s41579-018-0141-x, 10.1002/9781119282549.ch17, 10.3390/antibiotics11020200). Novelty of the manuscript with the detailed comparison with existing reviews has to be added to the introduction section.

2. The second major issue is the vague scope of this review. Even if section 2 is considered as the part of introduction, the reasoning for choosing the main sections remains to be elusive. For example, drug combination section (3.1) includes also an example of covalent dual drug adduct, but provide the sole example. The combination of gentamicin and ampicillin is mentioned in other section (3.2). There are some sections, devoted to targeted delivery, using liposomes (3.4) and sideromycins (5), whereas other approaches to the targeted delivery are not mentioned (e.g. immunoconjugates, aptamers, other biomolecules, etc). Therefore, clear structure and scope of the manuscript has to be provided. The manuscript has to be reorganized considering the scope and the novelty, clearly stated in the introduction

Minor

1. Most of the subsections are short and can be merged into single section

2. There is no abstract in the manuscript

3. Section titles end with a colon

Author Response

According to the comments of the referee

1) reviews dealing with  the main topic of this review or on specific sections have been mentionned ( references 4-7). We have also justified the novelty of our review  focusing on papers reporting the new “addressing strategies” to ensure efficient delivery of antibiotics to internal targets: for instance, the use of “biological cargo” that parasite the uptake transport of specific nutrients since iron, or sugars or liposomes. These Trojan horses have been described to illustrate this original way for repurposing antibiotics that are inactive against Gram-negative bacterial strains. By this way, the cargo moiety ensures the intracellular penetration without exhibiting additional antibacterial activity per se, a key advantage that prevents a future emergence of resistance mechanisms. Thus, this review will not deal with immunoconjugates as well as aptamers strategies since numerous reviews have been devoted to this strategy  (references 8-11)

The manuscript has been reorganized with a part devoted to siderophores (subsections have been merged into  a single section) 

An abstract has been added according to the recommendation of the referee

Colon was removed for Section titles end  for clarity.

Reviewer 3 Report

It's an interesting review of the hot topic of bacterial resistance. The text is well-written and sufficiently detailed. The pictures are nicely designed. I think that the manuscript would be accepted in its present form

Author Response

We thank the referee for his positive comments. 

Round 2

Reviewer 2 Report

The problem with the manuscript was not solved in the revised version. There are numerous reviews on the exactly the same topic or very close one, not even mentioned in this manuscript. For example, https://doi.org/10.1021/acs.jmedchem.8b00522; https://doi.org/10.1002/med.21588; https://doi.org/10.4155/fmc-2020-0065; https://doi.org/10.3390/antibiotics11030412.

To meet the criteria of novelty for the review, the following has to be done:

1. All of the reviews on the theme (not only those listed above) has to be mentioned in the manuscript

2. The experimental works, cited in the previous reviews, are to be excluded from the manuscript.

Author Response

Dear editor

Please find enclosed our revised manuscript ijms-2110856 entitled "Rejuvenating the activity of usual antibiotics on resistant Gram-negative bacteria, recent issues and perspectives”.

According to the recommendation of the referee we have added references (see the list at the end of this letter) dealing with the same topic. We want to underline that this review article presents a parallel between a smart drug design of antimicrobial agents and a specific delivery located in the cytoplasm or in the periplasm of the bacteria which is an original point of view with respect to all the previous reviews in literature.

Taking into accounts that (i) for all the two other referees “no change was necessary concerning the cited experimental works” we do not subscribe to the referee’s suggestion for the removal of the experimental works (not previously suggested by this referee) cited since (ii) these methodologies greatly contribute to the understanding of the future strategies involved and discussed in the article.

Hoping that this manuscript will be now suitable for publication in International Journal of Molecular Sciences.Thank you for your and the reviewer’s consideration. Very truly yours.  

  1. M. Brunel

References

Schalk IJ. A Trojan-Horse Strategy Including a Bacterial Suicide Action for the Efficient Use of a Specific Gram-Positive Antibiotic on Gram-Negative Bacteria. J Med Chem. 2018 May 10;61(9):3842-3844

Pham TN, Loupias P, Dassonville-Klimpt A, Sonnet P. Drug delivery systems designed to overcome antimicrobial resistance. Med Res Rev. 2019 Nov;39(6):2343-2396.

Skwarczynski, M.; Bashiri, S.; Yuan, Y.; Ziora, Z.M.; Nabil, O.; Masuda, K.; Khongkow, M.; Rimsueb, N.; Cabral, H.; Ruktanonchai, U.; et al. Antimicrobial Activity Enhancers: Towards Smart Delivery of Antimicrobial Agents. Antibiotics 2022, 11, 412.

Dassonville-Klimpt A, Sonnet P. Advances in 'Trojan horse' strategies in antibiotic delivery systems. Future Med Chem. 2020 Jun;12(11):983-986. 

Collins SM, Brown AC. Bacterial Outer Membrane Vesicles as Antibiotic Delivery Vehicles. Front Immunol. 2021 Sep 20;12:733064. doi: 10.3389/fimmu.2021.733064. PMID: 34616401; PMCID: PMC8488215.

Stebbins ND, Ouimet MA, Uhrich KE. Antibiotic-containing polymers for localized, sustained drug delivery. Adv Drug Deliv Rev. 2014 Nov 30;78:77-87. doi: 10.1016/j.addr.2014.04.006. Epub 2014 Apr 18. PMID: 24751888; PMCID: PMC4201908.

Brooks BD, Brooks AE. Therapeutic strategies to combat antibiotic resistance. Adv Drug Deliv Rev. 2014 Nov 30;78:14-27. doi: 10.1016/j.addr.2014.10.027. Epub 2014 Oct 28. PMID: 25450262.

Cheng AV, Wuest WM. Signed, Sealed, Delivered: Conjugate and Prodrug Strategies as Targeted Delivery Vectors for Antibiotics. ACS Infect Dis. 2019 Jun 14;5(6):816-828. doi: 10.1021/acsinfecdis.9b00019. Epub 2019 Apr 10. PMID: 30969100; PMCID: PMC6570538.

Round 3

Reviewer 2 Report

The authors claim that the manuscript has an original point of view on the literature, summarized in previous reviews. Nonetheless, there is still no comparison with the reviews in the introduction section, clearly pointing out this point of view and comparing it with previous reports on the same topic.

The article can be suitable for publication if  complete description of this original point of view will be added in the introduction section. The introduction should provide clear understanding of the angle, scope and novelty of the review. Now there is only the statement "review will not deal with immunoconjugates as well as aptamers strategies since numerous reviews have been devoted to this strategy". The review is focused on the "new strategies", but there it remains unclear, how experimental works are evaluated for inclusion, whereas there are both rather old experimental reports (5 years and older) and those previously included in other review articles.

Author Response

Dear editor

Please find enclosed our revised manuscript ijms-2110856 entitled "Rejuvenating the activity of usual antibiotics on resistant Gram-negative bacteria, recent issues and perspectives”.

According to the recommendation of the referee we have modified

- the introduction section to detail the key points addressed in this review

- and the conclusion to include a parallel between a smart drug design of antimicrobial agents and the specific delivery located in the cytoplasm or in the periplasm of the bacteria depending on “Cargo” used which is an original point of view with respect to all the previous reviews in literature. A recent reference dealing with the maltocargo strategy was also added.

Hoping that this manuscript will be now suitable for publication in International Journal of Molecular Sciences.Thank you for your and the reviewer’s consideration. Very truly yours.  

  1. M. Brunel